# Tibial Tuberosity–Tibial Intercondylar Midpoint Distance Can Be Interchangeably Measured on Axial CT and MRI: Retrospective Cross-Sectional Comparative Study

**DOI:** 10.3390/medicina61020348

**Published:** 2025-02-17

**Authors:** Dinko Nizić, Marko Šimunović, Jure Serdar, Josip Vlaić, Mario Josipović, Ivan Levaj, Igor Ivić-Hofman, Mislav Jelić

**Affiliations:** 1Special Hospital Agram, Trnjanska Cesta 108, 10000 Zagreb, Croatia; dinko.nizic@gmail.com; 2Special Hospital Medikol, Ulica Dragutina Mandla 7, 10000 Zagreb, Croatia; 3Department of Orthopaedic Surgery, School of Medicine, University Hospital Center Zagreb, University of Zagreb, Šalata 6–7, 10000 Zagreb, Croatia; jure.serdar@yahoo.com (J.S.); mariojosipovic@gmail.com (M.J.); ivanlevaj@gmail.com (I.L.); ortopedija@kbc-zagreb.hr (M.J.); 4Division of Paediatric Orthopaedic Surgery, Children’s Hospital Zagreb, Ulica Vjekoslava Klaića 16, 10000 Zagreb, Croatia; jvlaic@yaho.co.uk; 5Public Health Teaching Institute of Brod–Posavina County, Vladimira Nazora 2, 35000 Slavonski Brod, Croatia; javno-zdravstvo@zzjzbpz.hr

**Keywords:** CT, MRI, patellar instability, tibial intercondylar midpoint, tibial rotation, tibial pericondylar rectangle, TT–TG distance, TT–TIM distance

## Abstract

*Background and Objectives*: It is unknown whether the tibial tuberosity–tibial intercondylar midpoint (TT–TIM) distance can be interchangeably measured on axial computed tomography (CT) and magnetic resonance imaging (MRI). The objective of this retrospective cross-sectional comparative study was to evaluate the intermethod agreement of the TT–TIM distance on axial CT and MRI and its bias towards tibial rotation (TR), age, sex, and body side. *Materials and Methods*: On axial CT and MRI of 15 consecutive knee pairs where each pair belonged to the same patient with no pathology affecting the tibial circumference and tibial tuberosity, TT–TIM distance and TR were measured by two blinded radiologists at 2-week intervals. Upon checking the symmetry of distributions (Shapiro–Wilk test), differences between matched knee pairs (Wilcoxon signed-rank test), intermethod (Bland–Altman plot) and interrater agreement (intraclass correlation coefficient [ICC]), and correlations (Spearman rank correlation) were assessed. *Results*: The mean intermethod difference in TT–TIM distance was not statistically significant (−0.4 mm [−1.82, 0.96]; *p* = 0.52). The TT–TIM distance did not differ between knee pairs (*p* = 0.68), its interrater agreement was almost perfect (ICC > 0.81), and no bias towards TR (*p* > 0.66), age (*p* > 0.14), sex (*p* = 0.66), and body side (*p* > 0.37) was found. *Conclusions*: The TT–TIM distance can be interchangeably measured on axial CT and MRI with almost perfect interrater agreement, unbiased towards TR, age, sex, and body side.

## 1. Introduction

Patellar instability is a frequent occurrence in the orthopedic practice. The incidence of patellar dislocations in 100,000 people is 5.8 in the general population, 29 in 10 to 17 year olds, and as high as 49 in young women engaged in physical activities [1,2,3]. Primary patellar dislocations are mostly treated conservatively, whereas recurrent dislocations would probably need operative treatment, in which case imaging of the alignment of extensor mechanism of the knee has an important role in the pre-operative work-flow [4]. Nevertheless, this represents a real diagnostic challenge due to the presence of often unreliable imaging results.

The misalignment of the extensor mechanism of the knee in the frontal plane is partly due to excessive lateral distance between the tibial tuberosity and the mechanical axis of the lower limb (MALL). A standard imaging test for measuring this offset is the tibial tuberosity–trochlear groove (TT–TG) distance [5,6]. Recently, a new tibial tuberosity–tibial intercondylar midpoint (TT–TIM) distance has been proposed as an alternative [7]. Since the TT–TIM distance is measured solely by computed tomography (CT), however, it is unknown whether its measurements differ by magnetic resonance imaging (MRI).

The primary objective of this study was to evaluate the intermethod agreement of the TT–TIM distance on axial CT and MRI. The secondary objective of this study was to evaluate the bias of the TT–TIM distance towards tibial rotation (TR), age, sex, and body side.

## 2. Materials and Methods

After approval of the Ethical Committee (class: 8.1-20/219-4, No.: 02/21 AG, 7 December 2020), a retrospective cross-sectional comparative study was undertaken. The Strengthening the Reporting of Observational Studies in Epidemiology (STROBE) guidelines were followed where appropriate [8].

All measurements were made in the freely available RadiAnt DICOM Viewer (version 2020.2. 32-bit; Medixant, Poznań. Poland) by two radiologists at 2-week intervals who were blinded to each other’s measurements. For CT, patients were imaged according to the CT protocol for patellofemoral instability (PFI), on a 16-row Siemens CT scanner (Siemens Medical Systems, Erlangen, Germany), with patients in the supine position, with legs extended, feet close together and the quadriceps muscles relaxed. The technical parameters were as follows: 40–51 mA, 130 kV, slice thickness 2 mm of the bone (width = 1500, mean = 450) and soft tissue windows (width = 450, mean = 45). For MRI, patients were recorded on a 1.5T Siemens MAGNETOM Avanto MR scanner (Siemens Medical Systems, Erlangen, Germany), and we used European Society of Skeletal Radiology Sports Subcommittee recommendations: Ax Int FS, FOV–16 cm, slice 4 mm, time to echo in milliseconds 40–50, matrix 256 × 256. The protocols were applied consistently to all patients.

The TT–TIM distance was the distance between the tibial intercondylar midpoint (TIM), which was the intersection of diagonals of a rectangle drawn around the tibial circumference, the tibial pericondylar rectangle (TPR), and the tibial tuberosity midpoint (TTM) of its most anterior bony part (Figure 1 and Figure 2) [7], with no pathology affecting the tibial circumference at its proximal level (for example fractures or tumors), as well as tibial tuberosity (for example fractures, tumors, osteotomy, and Osgood–Schlatter disease). In other words, any pathology including the measurement areas was excluded so it could not affect the measurement of the TT–TIM distance.

Now, planes, lines, and points are theoretical concepts. They were invented in an attempt to understand the biomechanical intricacy of the human body. According to the common orthopedic doctrine, we imagine MALL is a line passing through the midpoints of the hip, the knee, and the ankle in the frontal plane [9]. It seems to be not merely but an almost vertical line variably deviated from the true vertical for about 3° [10]. We can likely conceptualize the axis of the tibial tuberosity being a more or less vertical line in the frontal plane. In theory, these two lines overlap. In practice, they are surely close as tibial tuberosity is close to MALL in healthy knees [11]. If it were placed too laterally, though, the patella would grow chronically unstable. The goal of an imaging test is, therefore, to define the distance between these two lines and conclude whether they are too far apart so that an orthopedic surgeon might try to bring them closer together.

A line in the frontal plane, however, equals a point in the axial plane. CT and MRI examinations are performed in the axial plane, in what we might call a traditional default setting (discretely summarized in the well-known acronym: Computed Axial Tomography or CAT scan). In that type of setting, we are therefore measuring the distance between two points: the bony relative midpoint of the knee (where the MALL is allegedly passing through) and the bony midpoint of the tibial tuberosity (where the forces of the general quadriceps vector and thus the extensor mechanism of the knee converge). In measuring the TT–TIM distance, the former is represented by TIM and the latter by TTM of its most anterior bony part [7].

On the other hand, TR is an angle between the posterior intercondylar line of the tibia, connecting the two most posterior points of the proximal part of the bone, and the horizontal line, resulting in positive (external) or negative (internal) values [7,12].

According to the review of orthopedic charts, a recurrent patellar instability defined as at least two non-traumatic patellar dislocations was unambiguously reported in 73% (11/15) of consecutive knee pairs, with valgus deformity in 33% (5/15) of the total, whereas it was likely in the rest but not explicitly stated and either ambiguous or incomplete. However, clinical data on PFI were irrelevant for the study.

### Statistical Analysis

To detect a large (i.e., clinically significant) effect size whilst taking into account 2-sided *p* values with a 5% level of statistical significance [6] and 80% level of statistical power, a sample of 15 consecutive knee pairs (N = 30), where each pair belonged to the same patient, was required as calculated in the freely available G*Power software (version 3.1.9.2.; Faul F.: Kiel Universität, Kiel, Germany).

Mean values were reported together with the typical measure of data dispersion, the standard deviation (SD) (mean ± SD). Upon checking the symmetry of distributions (Shapiro–Wilk test) [13], differences between matched knee pairs (Wilcoxon signed-rank test), intermethod (Bland–Altman plot) [14] and interrater agreement (intraclass correlation coefficient [ICC]), and correlations (Spearman rank correlation coefficient, r_s_) were assessed [15].

## 3. Results

Excluding age (*p* = 0.004), all distributions were symmetric (*p* = 0.28 to 0.90). The median age was 18 years (range 15 to 66). The mean values of the TT–TIM distance were 13.4 ± 2.8 mm on CT, and 13.8 ± 3.4 mm on MRI (Table 1). The mean difference in the TT–TIM distance on CT and MRI was not statistically significant (−0.4 mm [95% CI: −1.82, 0.96], *p* = 0.52) (Figure 3). The TT–TIM distance did not differ between knee pairs (*p* = 0.68), and the interrater agreement was almost perfect (0.97 [0.92, 0.99] on CT, and 0.98 [0.95, 0.99] on MRI).

The mean values of TR were −5.3 ± 5.9° on CT and −7.3 ± 8.0° on MRI (Table 1), with corresponding ICCs of 0.99 (0.98, 1.00) on CT and 0.99 (0.99, 1.00) on MRI. The intermethod TR did not differ (*p* = 0.60) nor influence the measurement of the TT–TIM distance on CT (r_s_ = −0.02 [−0.53, 0.50], *p* = 0.94) and MRI (r_s_ = −0.13 [−0.60, 0.41], *p* = 0.66). The TT–TIM distance was unbiased towards age (*p* > 0.14), sex (*p* = 0.66), and body side (*p* > 0.37) (Table 2).

## 4. Discussion

The most important finding of this study is that the TT–TIM distance can be interchangeably measured on axial CT and MRI with almost perfect interrater agreement, unbiased towards TR, age, sex, and body side.

The standard imaging test for PFI imaging, the TT–TG distance, has numerous and well-known disadvantages, three of which are important with regard to the objectives of our study [7]. First, the TT–TG distance has a relatively high SD (around 6 mm or higher) which means it is imprecise (the lower the precision, the higher the chance the same patient could be categorized as positive or negative instead of repeatedly being categorized the same) [7]. Second, it is age-dependent [7,16,17,18]. Third, the values of the TT–TG distance are different on CT and MRI [19,20,21]. Our study has shown TT–TIM distance was much more precise (2.8 mm on CT and 3.4 mm on MRI, hence a difference around 3 mm), as well as age- and modality-independent, hence more convenient for practical use. What is more, SDs, given generally in biostatistics, are considered lower in the population, so one could speculate that the population’s SD of the TT–TIM distance may be lower than 3 mm. In addition, the type of MRI sequence is likely irrelevant, as long as the measurement of the TT–TIM distance is performed on an axial image because bony landmarks, required for the measurement (the tibial circumference on the »headless cut« producing the TPR with its diagonals intersecting in the tibial intercondylar midpoint and the most anterior bony part of the tibial tuberosity on the »prominent cut«), are easily recognizable (Figure 2). Other advantages of the TT–TIM distance, such as independence of body side, femoral trochlear dysplasia, and knee rotation, among others, including age- and TR-independence as confirmed in our study, have been previously documented [7]. Clearly, a more precise measurement, independent of various confounders, may render the PFI imaging more reliable and hence beneficial to both patients and imaging professionals.

So far, the TT–TIM distance has been described on axial CT imaging [7]. Although CT is superior to MRI vis à vis contouring bony landmarks [22], as well as time– and cost–benefit issues [23], with its effective knee dose of ionizing radiation far lower than one would care to presume [24], MRI is nowadays extensively used in this imaging venue [25]. It would be, therefore, quite practical if one would be able to reliably measure the TT–TIM distance on MRI. Apart from practicality, MRI would remove the exposure of the patient’s knee to ionizing radiation while giving a more detailed insight into any concomitant soft-tissue lesion of the knee joint, hence providing an orthopedic surgeon with more extensive diagnostic information during a surgical assessment.

The results at hand support the independence of the TT–TIM distance from TR, age, body side, and even sex, where the latter is doubtful [7] and far-fetched due to the unintentional preponderance (93%) of female knee pairs in our cohort. It should be noted that the TT–TIM distance is measured with respect to the axial plane of the tibia rather than that of the CT/MRI scanner. This might be why TR does not affect its values, unlike, for example, TT–TG and, to lesser extent, the tibial tuberosity–posterior cruciate ligament (TT–PCL) distance [26].

The measurement of the TT–TIM distance itself was practical. Finding an appropriate CT and MRI image to define the TIM was straightforward: we found the first image on which the top of the fibular head was no longer visible (the »headless cut«) and drew a tibial pericondylar rectangle (TPR), and the TIM was the intersection of its diagonals. Then, we drew a vertical line through it to project the point along that line and copied the line to all images of the series. We found this step to be very user-friendly, both on CT and MRI. What is more, TPR accounts for the condylar circumference of an individual tibia. On the one hand, the sex bias of tibial morphology may or may not have brought about sex bias of the TT–TIM distance [7,27]. On the other hand—given that sex is an individual trait—its bias makes the TT–TIM distance a personal measure. Finding an appropriate CT and MRI image to define the TTM was a bit trickier: we scrolled down the images of the tibia and located the last image *before* the tibial tuberosity started to “move” in the *posterior* direction (the »prominent cut«), and the TTM was the midpoint of the tibial tuberosity dome. This was our way to fine-tune the tip of the tibial tuberosity. Theoretically, this step might be a source of interrater disagreement and hence a possible precision deficit, needing further standardization, perhaps more so on MRI. The same suggestion would go for TT–TG and TT–PCL distances, though, as well as other similar measurements using tibial tuberosity as a bony landmark. Still, in spite of this potential variability, the interrater agreement was almost perfect (>0.81). Such a favorable correspondence was perhaps due to the standardized definition of the TIM but also the skill and know-how of the raters.

We would like to emphasize the value of TT–TIM distance in practice should be expressed as an integer value, a whole number on a millimeter scale. Then, mathematically speaking, a difference of 1 mm would constitute a true difference (for example, 12 mm would be different from 13 mm). This, in turn, would imply that in order to be regarded as essentially the same, the mean difference in CT and MRI measurements of the TT–TIM distance should be less than 1 mm. Given that such a strict frame of reference might be justified by the narrow interval of the TT–TIM’s normative values, and although the mean difference in our study (−0.4 mm) satisfied this criterion—yet should not be commented upon as its 95% CI contained zero (−1.82, 0.96), indicating a chance result—it is certainly open for future studies to more thoroughly elucidate whether this mathematical reasoning may or may not be a possible overkill in a clinical setting.

The primary objective of our study was to compare CT and MRI measurements of the TT–TIM distance irrespective of the clinical data while marginally probing into possible cofounders such as TR, age, sex, and body side. In the event of much larger sample sizes, however, we surmise that the difference would be statistically significant but nonetheless small, hence being clinically (i.e., practically) insignificant as indicated by the already narrow 95% CI (−1.82, 0.96). In other words, increasing the sample size might produce a statistically significant mean difference that is, in fact, negligible and yields no clinical merit [28]. We could certainly recognize a retrospective design in addition to female prevalence as some of the most evident limitations. Nonetheless, since neither a detailed orthopedic chart nor potential sex variability had any substantial relevance in fulfilling the primary objective of our study, these deficiencies might bear subsidiary weight. Perchance a thought of greater import is the fact that all participants were Caucasian due to local demographics, so the results of our study may not be directly applicable to different populations.

## 5. Conclusions

In conclusion, the TT–TIM distance can be interchangeably measured on axial CT and MRI with almost perfect interrater agreement, unbiased towards TR, age, sex, and body side.

## Figures and Tables

**Figure 1 medicina-61-00348-f001:**
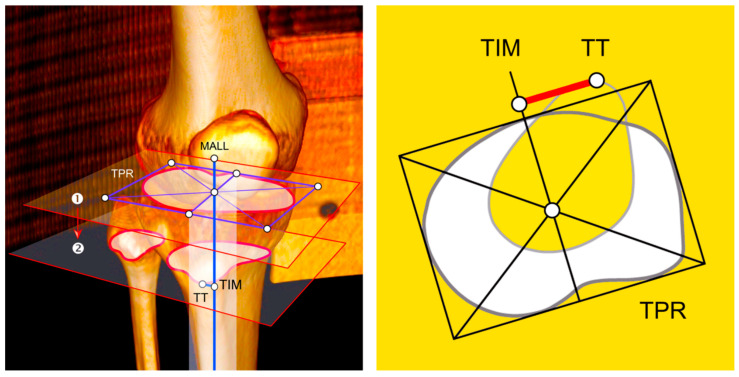
Illustration of the concept and measurement of the tibial tuberosity–tibial intercondylar midpoint (TT–TIM) distance. (**Left**) Illustration of the concept (3D CT reconstruction). (**Right**) Illustration of the measurement. ① First axial image where the top of the fibular head was no longer visible (»the headless cut«), ② axial image at the level of the most anterior bony part of tibial tuberosity (»the prominent cut«). MALL: mechanical axis of the lower limb, TIM: tibial intercondylar midpoint, TPR: tibial pericondylar rectangle, TT: tibial tuberosity.

**Figure 2 medicina-61-00348-f002:**
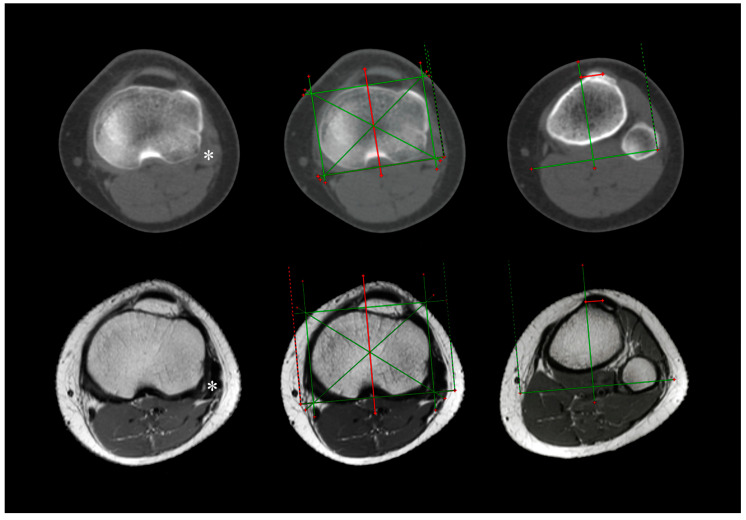
Axial CT (**upper half**) and MRI (**lower half**) measurement of the tibial tuberosity–tibial intercondylar midpoint (TT–TIM) distance on a left knee pair of the same patient. Asterisk (*) denotes the first axial image where the top of the fibular head was no longer visible (the »headless cut«).

**Figure 3 medicina-61-00348-f003:**
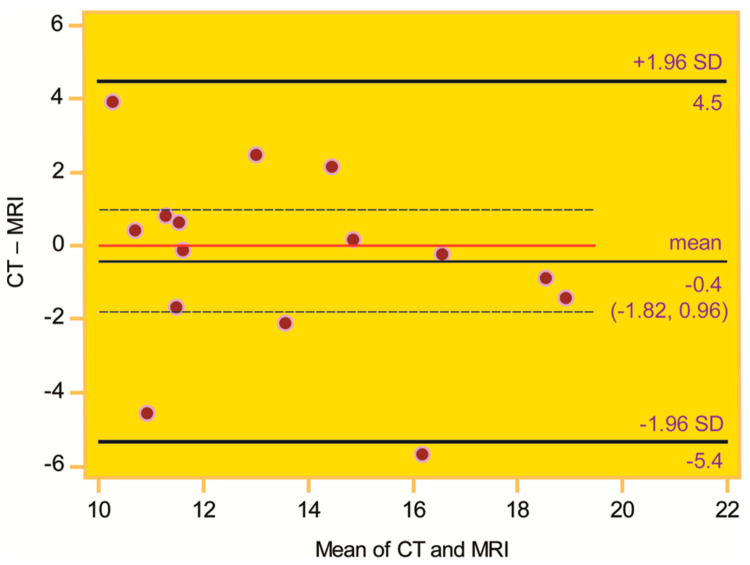
Intermethod agreement of CT and MRI measurements of the tibial tuberosity–tibial intercondylar midpoint (TT–TIM) distance on the Bland–Altman plot. The zero line (red) is within the 95% CI of the mean difference, indicating intermethod agreement.

**Table 1 medicina-61-00348-t001:** Measurements of the tibial tuberosity–tibial intercondylar midpoint (TT–TIM) distance and tibial rotation (TR) on CT and MRI.

N	Age	Sex	Side	TT–TIM Distance (mm)						TR (°)					
				CT (1)	CT (2)	CT (m)	MRI (1)	MRI (2)	MRI (m)	CT (1)	CT (2)	CT (m)	MRI (1)	MRI (2)	MRI (m)
1	16	F	R	12.4	12.6	12.5	14.5	14.8	14.7	−2.9	−3.5	−3.2	−20.3	−20.7	−20.5
2	18	F	R	15.7	17.2	16.5	16.8	16.6	16.7	5.3	5.9	5.6	−14.0	−15.0	−14.5
3			L	17.6	18.6	18.1	18.7	19.3	19.0	−1.0	−0.2	−0.6	−12.1	−12.4	−12.3
4	17	F	R	11.9	11.8	11.9	11.1	11.4	11.3	−7.3	−8.3	−7.8	3.2	3.0	3.1
5	15	F	L	11.6	12.9	12.3	8.5	8.3	8.4	−8.4	−8.5	−8.5	−5.1	−4.2	−4.7
6	54	F	R	10.4	11.4	10.9	10.8	10.2	10.5	−13.6	−14.7	−14.2	−6.9	−7.1	−7.0
7			L	11.2	12.2	11.7	11.6	10.2	10.9	−7.5	−7.7	−7.6	−12.1	−12.1	−12.1
8	17	F	R	16.0	15.1	15.6	13.3	13.5	13.4	−7.1	−6.4	−6.8	6.6	6.2	6.4
9	30	F	R	9.5	7.8	8.7	13.7	12.8	13.3	−7.8	−6.5	−7.2	−18.5	−19.4	−19.0
10	34	F	R	10.3	11.0	10.7	11.8	12.9	12.4	0.6	0.8	0.7	−5.8	−6.3	−6.1
11			L	11.8	11.3	11.6	11.3	12.1	11.7	1.3	1.1	1.2	1.5	1.7	1.6
12	60	F	R	13.1	13.6	13.4	19.6	18.5	19.1	1.1	0.0	0.6	−6.7	−6.5	−6.6
13	15	F	L	17.6	18.8	18.2	20.3	19.0	19.7	−14.6	−15.0	−14.8	−1.7	−2.6	−2.2
14	15	F	L	15.0	14.9	15.0	15.7	13.9	14.8	−10.2	−10.7	−10.5	−2.2	−2.2	−2.2
15	66	M	R	14.1	14.4	14.3	12.2	11.4	11.8	−6.5	−6.7	−6.6	−14.6	−13.7	−14.2

° = degrees; (1), (2) = raters; CT = computed tomography; F = female; L = left (knee); (m) = mean; M = male; mm = millimeters; MRI = magnetic resonance imaging; R = right (knee); TR = tibial rotation; TT–TIM = tibial tuberosity–tibial intercondylar midpoint (distance).

**Table 2 medicina-61-00348-t002:** Relationship of the tibial tuberosity–tibial intercondylar midpoint (TT–TIM) distance with age, sex, and body side.

Parameter	CT			MRI		
	r_s_	95% CI	*p* Value	r_s_	95% CI	*p* Value
Age	−0.41	−0.76, 0.14	0.14	−0.22	−0.66, 0.33	0.44
Sex	0.12	−0.42, 0.60	0.66	0.12	−0.60, 0.42	0.66
Body side	0.25	−0.30, 0.68	0.37	0.00	−0.51, 0.51	1.00

95% CI = 95% confidence interval; CT = computed tomography; MRI = magnetic resonance imaging; r_s_ = Spearman rank correlation coefficient.

## Data Availability

Data are contained within the article.

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
