# Peer review of "Tibial Tuberosity–Tibial Intercondylar Midpoint Distance Can Be Interchangeably Measured on Axial CT and MRI: Retrospective Cross-Sectional Comparative Study"

_medicina, 2025, doi:10.3390/medicina61020348_

Round 1
Reviewer 1 Report
Comments and Suggestions for Authors
In this study, the authors examined the interchangeability of measuring TT-TIM (the distance between tibial tuberosity and the midpoint of the tibial intercondyle) along axial CT and MRI, with an intention to assess their agreement with the biases potentially attributable to tibial rotation, age, sex, and body side. The manuscript is well-written and the methodology is sound; however, further clarity and depth in certain areas would enhance the manuscript's overall quality. Here are my comments:
Abstract
1. Please provide adequate information about the clinical implications of the findings. For example, mention briefly how the interchangeability or similarity of CT and MRI measurements might affect clinical decisions and plans for surgery.
Main text:
Introduction:
Further context that warrants its significance in the field of medicine should be provided. For example:
2. What is the ratio of occurrences of patellar instability and what might be the consequences if left untreated?
3. What challenges currently face diagnosis and management of this condition?
Although the Introduction mentions TT-TIM distance as a "new" option for TT-TG distance, a critical evaluation of the limitations that TT-TG distance has or how TT-TIM distance tries to solve these problems is lacking.
4. What about TT-TG distance has downsides, or shortcomings? The specific shortcomings could include sensitivity to rotational movement of the Tibia or variability in measurements.
5. How does TT-TIM solve those problems?
This is pointed out that CT has been used mainly in measuring TT-TIM distance, and it "remains unknown if it would differ in MRI." But no reason for the clinical relevance is provided. Add these to introduction:
6. Why would MRI be preferred in this context despite CT being the historically accepted modality?
7. What are the pros and cons of CT versus MRI in measuring TT-TIM distance?
The introduction has not quite clearly stated the goals of the study - primary and secondary - rather has concluded with a general statement regarding "intermethod agreement" and "the bias exerted by tibial rotation, age, sex, and body side." I suggest to clarify below questions in the objective part of introduction:
8. Is the primary purpose to compare TT–TIM distances, as measured on CT and MRI?
9. Are the secondary purposes to assess how tibial rotation, age, sex, and body side refine or bias these measurements?
The introduction cites only four references throughout, which are inadequate for the current study. The reason is:
10. There is no mention or discussion of earlier studies which compared CT to MRI for the same types of measurements (TT–TG distance, for example).
11. The manuscript does not engage with the broader literature on patellar instability or with the role of imaging in its management.
Expand the literature review to include more references.
Material and method:
12. Some justification for the retrospective design is lacking. Consider successful recovery of selection bias and incomplete data that are potential confounding factors.
13. For the reasons explained throughout this review, the term "case-control" is a misnomer; the authors did not compare cases with controls, but rather imaging via two imaging modalities (CT and MRI).
14. It is not clear how informed consent was obtained, particularly since the study was retrospective in nature. Were patients informed and consent waived by an ethics committee?
15. The article does not describe whether the study was conducted according to appropriate guidelines (STROBE for observational studies).
The inclusion and exclusion criteria are not described to the required detail. For instance:
16. What specific pathologies were excluded (e.g., fractures, tumors, Osgood-Schlatter disease)?
17. Were patients who had previous surgery or trauma to the knee included or excluded?
Further specificity regarding the imaging protocols is required. For example:
18. What were the technical parameters for CT (e.g., slice thickness, kVp, mAs)?
19. What MRI sequences were used (e.g., T1-weighted, T2-weighted), and what were the imaging parameters (e.g., slice thickness, field of view)?
20. It is not mentioned if the imaging protocols were done in a consistent fashion to all patients or if they varied.
21. While the "tibial pericondylar rectangle" (TPR) is detailed, it did not explain how it defined the tibial intercondylar midpoint (TIM). Further clarification on this is an important methodological point.
22. The authors didn't clarify how "most anterior part of the tibial tuberosity" was established and measured. This potentially results in variability between raters.
23. The article describes how tibial rotation was measured, but not in enough detail. However, how, for example, was the posterior intercondylar line and the horizontal line defined?
24. The authors have not presented any procedures for blinding of the observer in the manuscript. For example, were the radiologists blinded regarding the imaging modality (CT vs MRI) or to each other's measurements?
25. The manuscript does not state whether radiologists were trained/calibrated before the study to make standardized measurements.
26. The sample size of fifteen pairs of knees equaling N=30 was mentioned, but no rigorous justification was given for the authors' choice. Authors refer to statistical power without further explanation of the calculation used to arrive at that specific sample size.
27. It is asserted that "Data are contained within the article." However, this assertion is vague and insufficiently detailed with regard to the processes of accessing or reproducing the data.
Results:
28. The average difference of observed TR values with the hypothesis of roughly minus 2 degrees should be taken into consideration alongside the lack of clarity about reliance on the findings. The nature of clinical significance for the differential is also not explicitly stated. "Could the difference of 2 degrees influence the decision to perform surgery?" is for consideration.
29. The authors did not report data for range and dispersion related to TT-TIM measurements, which would have given the reader a better idea about the variability in measurements.
30. The authors did not comment upon the agreement intervals, as illustrated in the Bland-Altman plot; intervals are essential to understand the clinical acceptability pertaining to measurement interpretations.
31. The variability observed (-1.82 to 0.96 mm) is not commented on regarding its clinical acceptability.
32. The authors omitted ICC values which describe the reliability of TR, and it is a must to establish the reliability of the methodology used.
33. The authors did not discuss potential sources of variability between raters, for example, differences in their definition of the landmarks.
34. The authors did not provide raw correlation coefficients or effect sizes, and it makes it hard for the reader to assess the relative strength of the associations found.
35. Whether this lack of significance is due to the sample size being too small (N=30) or represents a true absence of bias is unclear.
36. There is no discussion of the TRs themselves, their variability, or their clinical importance.
37. There is no discussion as to whether these CT and MR TR discrepancies (-2.0°) can be viewed as clinically important.
38. The authors have not reported effect sizes or variability (e.g., SD, interquartile range) of any comparisons, so the magnitude of any differential values cannot be known.
39. There is no indication if an adjustment was made for multiple comparisons, which would make Type I errors more likely.
40. Please clarify what measurement interchangeability of CT versus MRI would imply about surgical planning or decision-making?
Discussion:
41. Discussion does not interpret the observed mean difference of –0.4 mm between CT and MRI measurements. Is such a difference negligible or clinically important in terms of surgical planning?
42. 42. It is unfortunate that there is no indication in the manuscript what a narrow 95% CI of –1.82, 0.96 for the mean difference between modalities means. Is this good news in terms of consistency in measurements across modalities?
The discussion grossly neglects the existing body of literature. For example:
43. 43. How do these findings compare to previous studies that have assessed CT and MRI in similar measurements (e.g., TT-TG distance)?
44. 44. How do the findings compare to previous studies? What explanations are given?
45. 45. The manuscript does not recognize the more extensive literature on patellar instability and the role of imaging in its treatment.
The discussion does not include concrete examples of how the above findings impact clinical decision-making or surgical planning, for instance:
46. The equivalent condition reflects a decrease in the redundancy of imaging studies.
47. This can be an example of how resources will be used or the effect on the further management of patients.
48. The manuscript does not mention possible adjustments in imaging protocols or guidelines in the light of the above findings.
49. It omits to clearly bring out the methodological strengths, including the use of unblinded raters, standardized measurement protocols, and inclusion of consecutive knee pairs.
The manuscript fails to describe sufficiently how these strengths inform the validity and reliability of the findings. The discussion too inadequately explores the implications of these limitations. In fact:
50. How does the retrospective design and the overrepresentation of females affect the generalizability of the findings?
51. A small sample size (N=30) may reduce the statistical power to detect bias or differences between modalities.
52. Other limitations not mentioned are the use of a single imaging software and landmark identification variability.
Comments on the Quality of English LanguageMinor grammatical and stylistic improvements would enhance readability and clarity.
Author Response
Dear reviewers
Thank you very much for your thorough and very detailed review.
We are sending you answers to your comments and questions in the attachment.

Reviewer 2 Report
Comments and Suggestions for Authors
Dear authors,
I had the opportunity to review your manuscript entitled "Tibial Tuberosity–Tibial Intercondylar Midpoint Distance Can Be Interchangeably Measured on Axial CT and MRI". Overall, the study is well-conceived, and the manuscript is easy to read, with a clear structure and a standardized approach to protocol and methodology. However, I have identified several areas where revisions would improve the manuscript's scientific quality and readability.
The study tackles a clinically relevant topic, offering valuable insights into the interchangeability of CT and MRI measurements. The protocol and methodology are well-standardized, ensuring reproducibility. The results are presented in a clear and organized manner.
Anyway, I had identified some aspects which can improve the scientific quality of your manuscript:
1. Introduction
The introduction is somewhat brief and would benefit from expansion to provide a more comprehensive background. I suggest addressing the following aspects:
- The clinical significance of the tibial tuberosity–tibial intercondylar midpoint distance in diagnosing and managing knee pathologies.
- The advantages and limitations of CT and MRI as imaging modalities.
Highlighting these points would offer a more robust context for the study and emphasize its relevance.
2. Materials and Methods
Lines 125–152, which describe methodological details, should be moved from the Discussion section to the Materials and Methods section. This change will enhance the clarity and logical flow of the manuscript.
3. Discussion
The discussion section needs better individualization and should include a comparative analysis with similar studies in the field. This will help situate your findings within the broader scientific literature and strengthen the interpretation of your results. Also, they could be further enriched by discussing the potential implications for clinical practice and future research directions.
4. Conclusions
The conclusions are consistent with the results and appropriately summarize the findings.
While the manuscript is well-constructed overall, I recommend its publication after major revisions to address the points outlined above. I look forward to the revised version of your manuscript.
Best regards.
Author Response

(The authors gave the same response as above.)

Round 2
Reviewer 1 Report
Comments and Suggestions for Authors
The revised manuscript represents a significant improvement and addresses most of the concerns from the review. The authors have further elaborated the methodological descriptions, explained key concepts, and expanded the discussion to give a much fuller clinical and scientific background to the study. However, some further opportunities remain to improve the manuscript, particularly in terms of providing more substantial justification for the sample size, being clearer about clinical acceptability interpretations, and discussing the wider implications for surgical planning and generalizability.
1. Although the authors have stated that the sample size was calculated using G*Power software, further details on the assumptions, such as effect size and alpha level, would strengthen the rationale for choosing a sample size of 15 knee pairs (N=30).
2. While the authors acknowledge the variability previously reported in TT-TIM measurements, it would strengthen their analysis to explicitly state the clinical acceptability of a mean difference of –0.4 mm (–1.82, 0.96); for example, comparing this figure to established thresholds for clinical significance for patellar instability.
3. The authors might comment on how the interchangeability of the CT and MRI measurements may impact surgical planning. That is, they might comment on whether the differences they found would change the threshold for performing a tibial tubercle osteotomy versus other interventions.
4. The authors comment on their population being predominantly white; however, they could have discussed more how their findings would apply to other populations, particularly those with differing anatomical variations or higher rates of patellar instability.
Author Response
Dear Editor(s), dear reviewers:
We hereby submit our responses to the reviewers’ questions. We would like to cordially thank the
reviewers for their valuable suggestions and for taking the time to write a detailed review of our
manuscript.
We submit reviewers’ questions and our answers in a table format: in the left column we copied and
pasted all the reviewer’s questions exactly as they were written, with their corresponding numbers
and our answers are given in the right column of the table. This makes the reading of this revision
much easier and faster to handle. Also, we have written our exact sentences that were changed or
added in the manuscript (in the revision of the manuscript, these sentences are written in blue and
underlined).
However, since many questions were with regard to the same topic or even the same theme within
the same topic, but were scattered throughout the revision, we have assembled these questions
regarding the same topic together and we have written our answers accordingly, to avoid unnecessary
redundancy of repeating the same things throughout our answers section.

Reviewer 2 Report
Comments and Suggestions for Authors
Dear authors,
I have carefully reviewed the revised version of your manuscript, along with your accompanying cover letter. In my assessment, the manuscript has undergone substantial improvements, particularly in terms of its scientific quality. It is evident that you have diligently addressed each of my previous suggestions, implementing them thoughtfully and effectively. As a result, the manuscript has significantly enhanced its clarity, rigor, and overall contribution to the field. Based on these improvements, I fully support its publication in its current form.
Best regards.
Author Response

(The authors gave the same response as above.)
